# Effectiveness of a Counseling Intervention to Increase Vaccination Uptake among Men Who Have Sex with Men during the Mpox Outbreak

**DOI:** 10.3390/vaccines12070751

**Published:** 2024-07-08

**Authors:** Matilde Ogliastro, Allegra Ferrari, Laura Sticchi, Alexander Domnich, Giacomo Zappa, Antonio Di Biagio, Elvira Massaro, Elisa Giribaldi, Andrea Orsi

**Affiliations:** 1Department of Health Sciences, University of Genoa, 16132 Genoa, Italy; allegraferrari@virgilio.it (A.F.); sticchi@unige.it (L.S.); antonio.dibiagio@hsanmartino.it (A.D.B.); elvss.m@gmail.com (E.M.); dott.elisa.giribaldi@gmail.com (E.G.); andrea.orsi@unige.it (A.O.); 2Hygiene Unit, San Martino Policlinico Hospital—IRCCS for Oncology and Neurosciences, 16132 Genoa, Italy; 3Local Health Unit 3 (ASL3), 16125 Genoa, Italy; giacomo.zappa@asl3.liguria.it; 4Infectious Diseases Unit, IRCCS Ospedale Policlinico San Martino, 16132 Genoa, Italy

**Keywords:** mpox, vaccination, MSM, counseling, HPV, hepatitis A, hepatitis B

## Abstract

Men who have sex with men (MSM) are disproportionately impacted by sexually transmitted infections (STIs), including HIV and those preventable through vaccination such as mpox, HPV, HAV, and HBV. A retrospective cohort study was conducted to evaluate the effectiveness of counseling provided during mpox vaccination on the uptake of other recommended vaccines (HPV, HAV, and HBV) and to identify associated factors. Relevant covariates such as nationality, age, HIV status, and use of PrEP were retrieved from electronic medical records. Vaccination status data were retrieved from the regional vaccination registry. Of the 330 participants, 98.8% were males and the mean age was 40.6 years (SD: 11.2). Following consultation, a statistically significant increase for both HPV (from 25.8% to 39.1%) and HAV (from 26.7% to 36.1%) was observed (*p* < 0.001). The multivariate analysis showed a significant negative association between the uptake of HPV and HBV vaccines and foreign nationality (aOR 0.25 (95%CI 0.08–0.69), *p* = 0.012; and aOR 0.31 (95%CI 0.11–0.81), *p* = 0.021). The HBV vaccine uptake was negatively associated with increasing age. Our results suggest that tailored counseling can effectively bridge the gap in vaccine acceptance among vulnerable populations, thereby improving overall public health outcomes.

## 1. Introduction

Human mpox [1] is a viral zoonosis caused by an enveloped double-stranded DNA virus belonging to the Orthopoxvirus genus within the *Poxviridae* family [2]. Mpox, endemic in Central and West Africa, occasionally causes outbreaks, mainly in the Democratic Republic of Congo. Until 2022, rare cases in non-endemic countries were usually linked to imports. In 2003, a cluster of 90 cases occurred in six U.S. states due to exotic animal importation from West Africa. This event marked the first human cases outside Africa, linked to contact with infected pets. Cases have since been reported in other countries. In 2022, widespread human-to-human transmission occurred in non-endemic areas, prompting the World Health Organization (WHO) to declare it a global health emergency in July 2022. Given the sustained decline in cases, the emergency was then declared over in May 2023 [3]. As of 30 April 2024, a total of 95,912 diagnosed cases of mpox have been reported worldwide, affecting 118 different countries and 111 of these nations had no historical record of mpox cases [4].

Historical risk factors for contracting the infection in African countries encompass living in forested regions (specifically near areas inhabited by squirrels), cohabiting in a household with documented monkeypox cases, being male, and being below 15 years of age [5]. However, in 2022, most patients diagnosed with mpox reported high-risk sexual behavior (e.g., sex with multiple partners) as a potential risk factor. Most cases have been identified in gay, bisexual, and other men who have sex with men (MSM) [5,6].

According to various studies [7,8,9], sexually active MSM are at greater risk of contracting sexually transmitted infections (STIs) including HIV [10] and those preventable through vaccinations, including human papillomavirus (HPV), hepatitis B (HBV), and hepatitis A (HAV) [11,12,13].

It is now well known from the scientific literature that human papillomavirus (HPV) is among the most commonly encountered sexually transmitted infections. Although it often progresses asymptomatically, HPV, particularly high-risk HPV genotypes, is also associated with the development of cancers. The most frequent of these is cervical cancer, but it also includes cancers of the vulva, anus, oropharyngeal tract, and penis. In 2018, at least 694,500 new cancer cases worldwide were attributable to HPV, including approximately 625,100 in women and 69,500 in men [14].

Men who have sex with men and transgender women experience a higher incidence of HPV compared to the general population [15]. A meta-analysis of 53 observational studies revealed that among MSM without HIV, the overall prevalence of anal HPV infection was 37%, with a prevalence of 64% for high-risk HPV types [16].

In 2019, according to the WHO, there were 286 million people with hepatitis B virus infection [11]. HBV infection remains a significant public health issue. Several studies have shown an independent correlation between high levels of viral DNA (>4 log(10) copies/mL or approximately 2000 international units/mL in individuals infected for more than 40 years) and an increased risk of developing cirrhosis and hepatocellular carcinoma (HCC) [17,18].

In Italy, HBV prevalence has decreased progressively over the past 50 years due to universal vaccination against HBV in force since 1991, improved socioeconomic conditions, better hygiene, reduction in household size, and awareness campaigns organized by the health authorities [19]. However, people who are unvaccinated with high-risk sex behavior remain at risk of acquiring HBV [20].

Globally, hepatitis A is estimated to affect 159 million people annually, resulting in 39,000 deaths [21]. Between 2016 and 2018 in Europe, there was an outbreak among men who have sex with men engaged in high-risk sexual behaviors [22]. In Milan, Italy, out of 353 identified cases, 172 were among MSM [23].

In Italy, the National Immunization Plan 2023–2025 recommends HPV, HBV, and HAV vaccinations to be provided free of charge for the MSM population [24].

These vaccines are also recommended for HIV-positive individuals [24]. People living with HIV are more vulnerable to severe manifestations of infections; for example, in the case of HPV, tumors and other complications associated with the virus are more frequently reported, especially among groups with low vaccination rates [25].

Additionally, the HBV vaccine is recommended for people who engage in sex work, and both the HAV and HBV vaccine are recommended for people with a substance use disorder [24]. It should be noted that some patients diagnosed with mpox have reported attending group sex sessions involving the use of recreational drugs (chemsex) [26].

Currently, despite the inclusion of high-risk MSM among the preferential target groups for the HPV, HAV, and HBV vaccination campaigns in Italy over the last three decades, vaccination uptake in this population remains suboptimal. In fact, a recent systematic review published in 2021 highlights that, despite the overall high level of vaccine acceptability, both vaccine uptake and completion rates have fallen short of the targets predicted by cost-effectiveness modeling [27]. Vaccination, despite its proven effectiveness in public health, is increasingly met with skepticism and hesitancy, which poses risks such as reduced coverage and heightened disease outbreaks in developed countries [28]. In recent decades, the concept of “hard-to-reach populations” has emerged to describe groups where vaccination rates are incomplete or absent. Understanding the dynamics of vaccination coverage and implementing strategies to enhance access are crucial. This includes addressing challenges faced by LGBTQ+ communities, who are often hesitant to disclose personal information due to past discrimination, impacting their healthcare-seeking behaviors, including vaccination uptake. A specific study focused on the COVID-19 vaccine for this population conducted an in-depth analysis of adherence and refusal, detailing factors contributing to vaccine hesitancy [29]. Predominant obstacles include social stigma, discrimination, lack of access, and low prioritization in vaccination campaigns, all of which contribute to uncertainty regarding vaccines in this community. Additionally, some informed individuals may not access vaccinations due to a lack of trust in the benefits of vaccines and healthcare institutions, making them physically accessible but resistant to vaccination. This distrust, often based on misinformation, can be heightened by fears of discrimination or other reasons. Socioeconomic status is a significant factor negatively affecting vaccine acceptance, as shown in several studies [30,31]. Although clear explanations for this influence are not always provided, socioeconomic status is often cited alongside other factors such as distrust in vaccinations, low education levels, and access issues [31].

Among the facilitators identified in this review, targeted information and awareness campaigns have been recognized as crucial tools for improving vaccine acceptance and uptake [27]. This aspect has been extensively explored during the COVID-19 pandemic, and the existing literature can provide a solid foundation for our study, emphasizing the importance of clear and inclusive communication to address vaccine hesitancy and enhance vaccination access. For example, a 2023 study by Cervi et al. underscores the need to develop efficient communication tools in a context marked by the increasing digitalization of society, where the use of the Internet and digital technologies has revolutionized communication [32]. As the “infodemic,” defined by the WHO as the dissemination of misinformation, continues to impact communities, implementing communicative strategies at both individual and community levels remains crucial. It is essential to discuss how individuals, especially in high-risk communities such as MSM, obtain information about vaccines and how authorities communicate the importance of vaccination. In addition to mass communication aspects highlighted primarily during the COVID-19 pandemic, reference to parallel communicative methods such as personalized counseling is also pertinent, as the correct use of health counseling skills enables structured and personalized communication, which is crucial for addressing concerns and individual perceptions related to vaccines.

The primary objective of this study was to evaluate the effectiveness of a vaccination counseling intervention carried out during the administration of the mpox vaccine, by comparing vaccine uptake before and after the counseling. The educational approach stood out for its focus on welcoming, listening, and respecting individual decisions regarding vaccination. The goal was not to convince or persuade, but rather to facilitate an informed and autonomous vaccination choice.

The secondary objective of our study was to identify the factors influencing vaccine uptake in this specific population.

## 2. Materials and Methods

A retrospective cohort approach was employed, focusing on individuals who were vaccinated against mpox in the metropolitan area of Genoa, (capital of the Liguria region, Northwest Italy) between 1 August 2022 (beginning of the vaccination campaign) and 29 February 2024.

The considered vaccination clinics included two locations within the jurisdiction: a territorial clinic located within the Department of Territorial Prevention and an intra-hospital clinic affiliated with the Department of Hygiene at the IRCCS Ospedale Policlinico San Martino in Genoa.

During the vaccination process, each patient received counseling from qualified personnel affiliated with the Department of Hygiene regarding the benefits and risks of recommended vaccinations for their specific risk category. This educational approach was based on principles outlined in the document “Counselling in ambito vaccinale” from the Italian National Institute of Health (ISS) on the importance of counseling in the context of vaccination, emphasizing active listening, understanding individual concerns, and supporting autonomous and informed decision making. The counseling was conducted by the same operators who administered the mpox vaccination, after consulting the patient’s vaccination booklet and collecting pre-vaccination medical history. Information regarding other recommended vaccines was provided not for co-administration, but to offer all necessary details for scheduling an appointment at an in-hospital vaccination clinic if the patient was interested. According to ministerial guidelines [15], patients had voluntary access to the clinic or were directed to do so in collaboration with the Department of Infectious and Tropical Diseases of IRCCS Ospedale Policlinico San Martino.

The regional vaccination registry provided by ASL3 (Local Health Authority, Azienda sanitaria Locale) was consulted for each patient before and after the administration of the mpox, to verify the vaccination status for HPV, HBV, and HAV.

Relevant covariates such as nationality, age, HIV status, and the use of PrEP were retrieved from electronic medical records. The data were extracted from the portal at the conclusion of the study period in March 2024 and compiled into a database.

Categorical data were presented as proportions and percentages along with Clopper–Pearson’s exact 95% confidence intervals (CIs), while continuous variables were reported as means ± standard deviations (SDs). The exact binomial test was applied to compare vaccination uptake before and after the intervention. To investigate factors associated with vaccination uptake, both univariable and multivariable logistic regression analyses were conducted, reporting odds ratios (ORs) and adjusted ORs (aORs), respectively. Variables with a significance level of *p* < 0.025 in the univariable analysis were included in the multivariable models. Statistical analyses were performed using R version 4.1.0 (R Foundation for Statistical Computing; Vienna, Austria).

This study was conducted within the PrE(P)VENZIONE 2023 project, and the protocol was approved by the Ethics Committee of the Liguria region (Ref. 273/2023—DB id 13210).

## 3. Results

A total of 330 individuals were enrolled in this study.

Of the participants, 98.8% were male, while only 1.2% were female, and these were primarily laboratory personnel with potential direct exposure to Orthopoxviruses. The mean age of the participants was 40.6 years (SD: 11.2). They were divided into four age groups: 18–29 years, 30–39 years (which represented the majority of vaccinated individuals, with 39.1%), 40–49 years, and over 50 years old.

Out of the total vaccinated subjects, 86.7% were Italians, while 13.3% had a foreign nationality of origin. Regarding HIV status, 24.5% were living with HIV, 37% were not living with HIV at the time of mpox vaccine administration, and the remaining 38.5% had an unknown status. Regarding the use of pre-exposure prophylaxis (PrEP), which was considered either continuous or on-demand at least once in their lifetime, 11.8% of patients had used it, while 33.0% had never used it. For 55.2% of patients, this information was unknown. The characteristics of the population are summarized in Table 1.

Regarding the primary objective of the study, Figure 1 demonstrates an increase in vaccination uptake pre- and post-counseling for all the vaccinations studied. The increase was statistically significant (*p* < 0.001) for both HPV and HAV uptake.

Table 2 presents the univariate and multivariate analyses of the study population. For each vaccine (HPV, HAV, and HBV), the uptake was studied according to age, nationality, HIV status, and PrEP use.

In the univariate analysis, factors associated with vaccine uptake with *p* < 0.025 for all vaccines considered included nationality of origin, PrEP use, and HIV status. Age was a significant factor only for the HBV vaccine.

In the multivariate analysis, the post-intervention uptake of HPV and HBV vaccines was significantly negatively associated with being foreign-born (aOR 0.25 (95%CI 0.08–0.69), *p* = 0.012; and aOR 0.31 (95%CI 0.11–0.81), *p* = 0.021, respectively). Additionally, the HBV vaccine uptake was negatively associated with increasing age (aOR 0.95 (95%CI 0.91–0.98), *p* < 0.001).

## 4. Discussion

Following the initiation of the anti-mpox vaccination program and counseling in the metropolitan area of Genoa, vaccination uptake for other recommended vaccines increased, suggesting a positive impact on vaccination behavior. The implication of pre-counseling uptake vaccination is that initial vaccination coverage rates were notably low, particularly for HPV and HAV, at 25.8% and 26.7%, respectively, before counseling. For HBV, the coverage was slightly higher at 47%.

The increase in vaccination uptake underscores the significant role that physicians can play in vaccine campaigns aimed at preventing STDs, especially during epidemics. By proactively offering vaccinations and making them more accessible, physicians can significantly impact public health outcomes, as highlighted in other studies [33,34].

The counseling approach emphasizing respect, active listening, and tailored information has proven effective in our study. The most significant increase was observed for the anti-HPV vaccine, rising from 25.8% to 39.1%. These data align with figures reported in a 2020 meta-analysis of HPV vaccination among MSM in the United States, which indicate an average HPV vaccination uptake of 38% [15]. The uptake of the HBV and HAV vaccines also showed an increase.

However, it is important to note that despite the increase, a significant portion of individuals who were advised to get vaccinated did not follow through. This result may indicate that there are barriers on the supply side (overcome through vaccination counseling where the offer is extended and a facilitated path for vaccination is indicated) as well as on the demand side. Even when people are aware that vaccination is recommended and offered for free, they still may choose not to get vaccinated.

This finding aligns with a recent review published by Ozawa et al., which indicates greater disparities on the supply side than on the demand side [35]. Similarly, since data collection occurred in March 2024 and the study considered vaccinations up to 29 February 2024, it is possible that this study does not include some individuals who may get vaccinated in the future for the recommended vaccines, even if they have not yet done so. In our context, hard-to-reach populations for vaccination include those who may face discrimination based on nationality of origin, socioeconomic status (SES), sexual orientation, and gender identity [29]. Stigma and fear of discrimination often represent relevant barriers to accessing healthcare and prevention services within the LGBTQIA+ community [36]. A recent cross-sectional study conducted in China among 7538 MSM demonstrated that after concern of safety and side effects, concern of privacy was the most common factor associated with hesitancy (38.24%) [37].

To address the issue of privacy concerns, an effective strategy based on the collaboration between the infectious disease unit and the hygiene unit was implemented at IRCCS Ospedale Policlinico San Martino. In particular, by implementing an informative intervention that commenced with patient education during a visit for STIs at the infectious disease unit and continued with a consultation for anti-mpox vaccination within the hygiene unit, the healthcare providers’ team could support the patient’s decision-making process and enhance preventive services’ accessibility within the same hospital setting.

For all the vaccines examined—HPV, HAV, and HBV—the univariate analysis revealed a significantly negative association between being a foreign nationality of origin and vaccination uptake. This association was also shown in the multivariate analysis for HPV and HBV vaccines. This finding aligns with a recent review by Crawshaw et al., which examined vaccination uptake in 30 European Union countries between 2000 and 2021 for various types of vaccinations. The review identified several obstacles encountered by those who are foreign, including linguistic and legal barriers, as well as supply-side barriers stemming from the lack of targeted guidelines [38].

Indeed, it should be noted that while most of the healthcare providers of Policlinico San Martino could speak Italian and varying degrees of English as a second language, translation and interpretation services from other languages, as well as cultural mediation services, were not available at the time of the counseling.

With regards to age, a significantly negative association was observed between HBV vaccine uptake and increasing age, in both univariate and multivariate analyses. This specific trend for HBV vaccination may be explained by the fact that in Italy, since 1991, HBV vaccination has become mandatory for all newborns and for adolescents at the age of 12 [39].

This study also examined HIV status and adherence to other preventive strategies such as PrEP. A univariate analysis indicated that people living with HIV had a lower vaccine uptake compared to those with negative serology. However, after adjusting for relevant covariates, this difference was no longer observed, and is consistent with other studies where no statistically significant difference was found in vaccine uptake among people living with HIV [40]. Individuals who use PrEP exhibit a higher level of vaccination uptake in comparison to those who are not utilizing PrEP, although this was not statistically significant at the multivariate analysis. This finding resonates with observations from previous published studies, which have consistently indicated a heightened level of vaccine adherence within this demographic [34]. This may be explained by both risk- and awareness-related factors. In fact, while studies have found that about 40% of patients infected with mpox are people taking PrEP [26], these patients have been more thoroughly informed about STIs prevention during recurrent specialist visits at infectious disease units, as is necessary for the prescription of the prophylactic therapy.

Limitations to this study include a considerable lack of data on HIV status (38.5%) and PrEP use (55.2%). Additionally, information on HIV diagnosis date and therapy adherence was not available. The vaccination status of people residing outside of the Liguria region was also unknown.

Finally, although transgender people are disproportionately impacted by HIV and other STIs [41], no information on gender identity was available and our analyses were solely based on sex assigned at birth.

In conclusion, while there has been an increase in vaccination uptake following the initiation of the anti-mpox vaccination program, compliance for recommended vaccinations in MSM at increased risk remains somewhat limited.

The recent WHO ‘Strategic framework for enhancing prevention and control of mpox (2024–2027)’, which outlines a road map for health authorities to control mpox outbreaks worldwide, emphasizes the importance of integrating the efforts of all health programs, including immunization and clinical services, epidemiological surveillance, primary healthcare, sexual health services, and community-based programs [42].

Indeed, coordination among all partners remains essential to ensure a continued robust response for mpox and other STIs. These efforts must be approached from an intersectional perspective, recognizing that marginalized communities, such as people who identify as LGBTQIA+, people with lower socioeconomic status, people who are part of racial and ethnic minorities, people with a substance use disorder, and sex workers, face multifaceted barriers to accessing healthcare.

In this regard, it should be noted that this study is part of a two-step project (‘PrE(P)VENZIONE 2023’) with the goal of identifying disparities in access to health and preventive services among populations at risk for STIs. To complement the findings of this cohort study with a community perspective, a cross-sectional survey on factors associated with access and the utilization of health and preventive services among individuals affiliated with PrEP centers and LGBTQIA+ associations in Italy is currently being conducted [43].

This study demonstrates that the intervention significantly increased vaccine uptake. The counseling intervention effectively addressed the challenges posed by initial low uptake rates, thereby enhancing vaccination acceptance and coverage within the targeted community. For instance, it is known that individuals who get vaccinated for different types of vaccines are more likely to continue this behavior in the future, as indicated by the literature on vaccination history and its influence on future immunization prospects [44,45]. Future studies should explore additional correlations, especially for diseases like mpox. According to the motto “*Nothing about us without us*”, used to communicate the idea that no policy should be decided by any representative without the full and direct participation of members of the group affected by that policy [46], incorporating the perspective of the involved communities is essential to bridge the current knowledge gaps and ensure that all involved populations receive targeted care and information.

## Figures and Tables

**Figure 1 vaccines-12-00751-f001:**
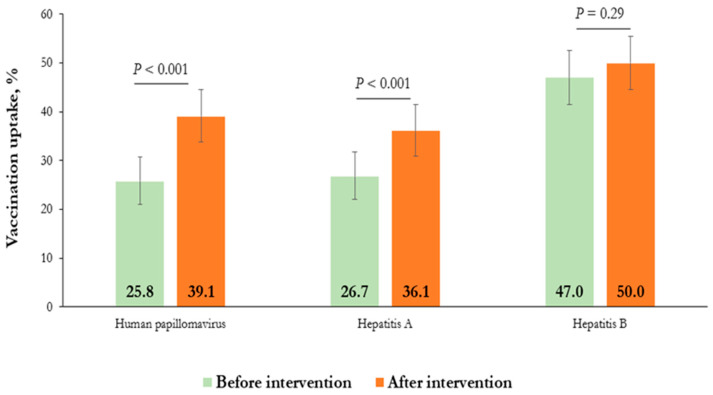
Effect of counseling intervention on vaccine uptake: comparative analysis before and after mpox vaccination.

**Table 1 vaccines-12-00751-t001:** Characteristics of the study population.

Variable	Level	% (n)
Sex	Female	1.2 (4)
	Male	98.8 (326)
Age, years	Mean (SD)	40.6 (11.2)
	18–29	16.4 (54)
	30–39	39.1 (129)
	40–49	23.9 (79)
	≥50	20.6 (68)
Foreign nationality of origin	Yes	13.3 (44)
	No	86.7 (286)
Living with HIV	Yes	24.5 (81)
	No	37.0 (122)
	Unknown	38.5 (127)
PrEP use	Yes	11.8 (39)
	No	33.0 (109)
	Unknown	55.2 (182)

**Table 2 vaccines-12-00751-t002:** Determinants of the post-intervention vaccination coverage according to the independent variables considered.

Variable		Vaccination Uptake, % (n)	OR (95% CI)	*p*	aOR (95% CI)	*p*
**Human papillomavirus (HPV)**						
Age, years	1-year increase	–	1.01 (0.99–1.03)	0.39	–	–
	18–29	42.6% (23)	Ref	Ref	–	–
	30–39	34.9% (45)	0.72 (0.38–1.39)	0.33	–	–
	40–49	38.0% (30)	0.83 (0.41–1.68)	0.59	–	–
	≥50	45.6% (31)	1.13 (0.55–2.33)	0.74	–	–
Nationality of origin	No	41.3% (118)	Ref	Ref	Ref	Ref
	Yes	25.0% (11)	0.47 (0.22–0.95)	0.043	0.25 (0.08–0.69)	0.012
Living with HIV	No	54.1% (66)	Ref	Ref	Ref	Ref
	Yes	34.6% (28)	0.45 (0.25–0.80)	0.007	0.44 (0.18–1.07)	0.072
PrEP use	No	40.0% (44)	Ref	Ref	Ref	Ref
	Yes	60.0% (27)	3.32 (1.55–7.47)	0.003	1.66 (0.59–4.74)	0.34
**Hepatitis A (HAV)**						
Age, years	1-year increase	–	0.99 (0.97–1.01)	0.54	–	–
	18–29	37.0% (20)	Ref	Ref	–	–
	30–39	38.0% (49)	1.04 (0.54–2.03)	0.90	–	–
	40–49	34.2% (27)	0.88 (0.43–1.83)	0.73	–	–
	≥50	33.8% (23)	0.87 (0.41–1.84)	0.71	–	–
Nationality of origin	No	39.2% (112)	Ref	Ref	Ref	Ref
	Yes	15.9% (7)	0.29 (0.12–0.64)	0.004	0.44 (0.14–1.21)	0.13
Living with HIV	No	45.9% (56)	Ref	Ref	Ref	Ref
	Yes	27.2% (22)	0.44 (0.24–0.80)	0.008	0.62 (0.25–1.57)	0.31
PrEP use	No	30.0% (33)	Ref	Ref	Ref	Ref
	Yes	52.5% (21)	2.69 (1.27–5.75)	0.010	1.73 (0.64–4.78)	0.28
**Hepatitis B (HBV)**						
Age, years	1-year increase	–	0.95 (0.92–0.97)	<0.001	0.95 (0.91–0.98)	0.001
	18–29	68.5% (37)	Ref	Ref	–	–
	30–39	58.1% (75)	0.64 (0.32–1.24)	0.19	–	–
	40–49	39.2% (31)	0.30 (0.14–0.61)	0.001	–	–
	≥50	32.4% (22)	0.22 (0.10–0.47)	<0.001	–	–
Nationality of origin	No	81.7% (152)	Ref	Ref	Ref	Ref
	Yes	29.5% (13)	0.37 (0.18–0.72)	0.005	0.31 (0.11–0.81)	0.021
Living with HIV	No	54.9% (67)	Ref	Ref	Ref	Ref
	Yes	38.3% (31)	0.51 (0.29–0.90)	0.021	0.83 (0.33–2.11)	0.70
PrEP use	No	40.9% (45)	Ref	Ref	Ref	Ref
	Yes	57.5% (23)	2.04 (0.98–4.36)	0.059	1.37 (0.49–3.88)	0.55

## Data Availability

The data presented in this study are available in this article.

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
