# Peer review of "Effectiveness of a Counseling Intervention to Increase Vaccination Uptake among Men Who Have Sex with Men during the Mpox Outbreak"

_vaccines, 2024, doi:10.3390/vaccines12070751_

Round 1
Reviewer 1 Report
Comments and Suggestions for Authors
Thanks for giving me the chance to read this interesting paper.
The manuscript starts from the premise that Men who have sex with men are disproportionately impacted by sexually transmitted infections.
Accordingly the authors conduct a retrospective cohort study to evaluate the effectiveness of 18 counseling provided during mpox vaccination on the uptake of other recommended vaccines (HPV, 19 HAV, HBV) and to identify associated factors.
The authors conclude that effective communication is a crucial tool for enhancing awareness and inclusivity of vaccination campaigns targeted at high-risk communities.
As such this work has the potentiality to effectively contribute to both the general literature dealing with counseling and vaccines and the more specific stream of research interested in Men who have sex with men.
In addition the paper can be of great interest also for readers coming from the Social Sciences.
The paper is methodologically impeccable and its results are very interesting, however it lacks a fundamental part: a theoretical grounding.
In the Introduction authors simply introduce the readers to the specific disease without grounding the study.
Accordingly it is seminal to add a Theoretical Section that:
1) Introduce the topic of vaccine and vaccine hesitancy.
There is a wide literature on this topic and during Covid-19 there has been a boom in research.
These literature must be acknowledged.
See for instance:
Dubé, Eve, Caroline Laberge, Maryse Guay, Paul Bramadat, Réal Roy, and Julie A. Bettinger. "Vaccine hesitancy: an overview." Human vaccines & immunotherapeutics 9, no. 8 (2013): 1763-1773.
2) Since the author conclude that “effective communication is a crucial tool for enhancing awareness and inclusivity of vaccination campaigns targeted at high-risk communities”, it is seminal to discuss a) how people gather information about the vaccines b) how authorities communicate vaccines.
This last point has been widely analyzed for Covid-19. These literature can help grounding your study.
See:
https://www.doi.org/10.4185/RLCS-2023-1845
3) Since the author tackle the specific case of Men who have sex with men, they should analyze previous literature on this specific cohort
Once the Theoretical Section and the lit Review has been enriched, authors should allow their results to dialogue with previous studies on the same topic or similar topics pointing out similarities and differences
Comments on the Quality of English Languageno comment
Author Response
Comment: Thanks for giving me the chance to read this interesting paper. The manuscript starts from the premise that Men who have sex with men are disproportionately impacted by sexually transmitted infections. Accordingly the authors conduct a retrospective cohort study to evaluate the effectiveness of 18 counseling provided during mpox vaccination on the uptake of other recommended vaccines (HPV, 19 HAV, HBV) and to identify associated factors. The authors conclude that effective communication is a crucial tool for enhancing awareness and inclusivity of vaccination campaigns targeted at high-risk communities. As such this work has the potentiality to effectively contribute to both the general literature dealing with counseling and vaccines and the more specific stream of research interested in Men who have sex with men. In addition the paper can be of great interest also for readers coming from the Social Sciences. The paper is methodologically impeccable and its results are very interesting, however it lacks a fundamental part: a theoretical grounding.
Reply: Thank you for your thoughtful and insightful feedback on our paper. We are pleased to hear that you found the manuscript interesting and recognize its potential contribution to both general and specific streams of research. We appreciate your kind words about the methodological rigor and the relevance of our results. Your point about the need for a theoretical grounding is well taken, and we agree. As a result, we have proceeded with revisions to strengthen the theoretical framework underpinning our study. Thank you once again for your valuable comments and for taking the time to review our work.
Comment: In the Introduction authors simply introduce the readers to the specific disease without grounding the study. Accordingly it is seminal to add a Theoretical Section that:
1) Introduce the topic of vaccine and vaccine hesitancy. There is a wide literature on this topic and during Covid-19 there has been a boom in research. These literature must be acknowledged. See for instance:
Dubé, Eve, Caroline Laberge, Maryse Guay, Paul Bramadat, Réal Roy, and Julie A. Bettinger. "Vaccine hesitancy: an overview." Human vaccines & immunotherapeutics 9, no. 8 (2013): 1763-1773.
Reply: Thank you for your comment and for this insightful suggestion. We fully agree with you, and we have incorporated this aspect into our work. We have introduced a section in the introduction that discusses the topic of vaccines and vaccine hesitancy. Drawing on the extensive literature highlighted by COVID-19 research, we have sought to explore the determinants of vaccine hesitancy, particularly within the specific population addressed in our study.
Comment: 2) Since the author conclude that “effective communication is a crucial tool for enhancing awareness and inclusivity of vaccination campaigns targeted at high-risk communities”, it is seminal to discuss a) how people gather information about the vaccines b) how authorities communicate vaccines.
This last point has been widely analyzed for Covid-19. These literature can help grounding your study.
See: https://www.doi.org/10.4185/RLCS-2023-1845
Reply: Thank you for highlighting this important aspect. We agree that it is crucial to discuss how people gather information about vaccines and how authorities communicate this information, especially in the context of vaccination campaigns targeting high-risk communities. The insights from the literature on Covid-19 communication, as indicated in the article you provided (https://www.doi.org/10.4185/RLCS-2023-1845), have been very useful and we have incorporated them into our introduction.
Comment: 3) Since the author tackle the specific case of Men who have sex with men, they should analyze previous literature on this specific cohort
Reply: In added sections, we have also focused specifically on the population under study. Thank you for these important contributions.
Reviewer 2 Report
Comments and Suggestions for Authors
This paper presents the results of a small retrospective study investigating the effect of counselling on vaccination uptake among individuals receiving mpox vaccination. The methodological section is however missing important details. It is not clearly described what "before" and "after" counselling groups mean. Is it simply the vaccination status of the same people before and after counselling? How long after counselling were data extracted from the national vaccination registry?
The counselling should also be described in details. Was it standardised? What did it include? Was it done by the same healtcare workers that vaccinated against mpox or separately?
It should also be explained in details how and where the vaccines in question were offered. Were they offered to people at the same time as the mpox vaccination or would people have to make separate arrangements for the HPV, HAV and HBV vaccines?
While not clear, it is indicated that the HPV, HAV and HBV vaccines are offered for free to the subjects in the current study. It is therefore surprising that the counselling has such a low impact (less than 10% for HAV and HBV, a bit more for HPV). This is maybe the most surprising finding, namely that people in a risk group being counselled to get a vaccines and yet only few follows this advice. This dilemma is not mentioned and should be elaborated.
Author Response
Comment: This paper presents the results of a small retrospective study investigating the effect of counselling on vaccination uptake among individuals receiving mpox vaccination. The methodological section is however missing important details. It is not clearly described what "before" and "after" counselling groups mean. Is it simply the vaccination status of the same people before and after counselling? How long after counselling were data extracted from the national vaccination registry?
Reply: Thank you for your valuable feedback. We agree. We have made the suggested modifications to our manuscript. In the methods section, we clarified that the counseling was conducted by the same operators who administered the mpox vaccination, after consulting the patient's vaccination booklet and collecting pre-vaccination medical history. Additionally, we explained that information regarding other recommended vaccines was provided not for co-administration but to offer all necessary details for scheduling an appointment at an in-hospital vaccination clinic if the patient was interested. We also specified that data extraction occurred in March 2024.
Comment: The counselling should also be described in details. Was it standardised? What did it include? Was it done by the same healtcare workers that vaccinated against mpox or separately?
Reply: Thank you for your feedback. We have incorporated additional details about the counseling process into our introduction. Specifically, we have detailed the educational approach used in our study, which draws on counseling principles from the Italian National Institute of Health (ISS). This approach focused on welcoming, listening, and respecting individual decisions regarding vaccination, with the aim of facilitating informed and autonomous choices.
Comment: It should also be explained in details how and where the vaccines in question were offered. Were they offered to people at the same time as the mpox vaccination or would people have to make separate arrangements for the HPV, HAV and HBV vaccines?
Reply: Thank you for your feedback. We have clarified that the vaccines were offered at the same clinic by appointment, scheduled on different days from the mpox vaccination.
Comment: While not clear, it is indicated that the HPV, HAV and HBV vaccines are offered for free to the subjects in the current study. It is therefore surprising that the counselling has such a low impact (less than 10% for HAV and HBV, a bit more for HPV). This is maybe the most surprising finding, namely that people in a risk group being counselled to get a vaccines and yet only few follows this advice. This dilemma is not mentioned and should be elaborated.
Reply: Thank you, we agree with your interesting feedback. Therefore, we have specified the points you suggested. In the discussion section, we emphasized that although there was an increase in vaccination coverage, a significant portion of the population did not get vaccinated despite the recommendations. We acknowledged that the early data collection might mean some individuals will get vaccinated in the future. Furthermore, we expanded on the barriers on both the supply and demand sides that still exist, as you rightly advised.
Reviewer 3 Report
Comments and Suggestions for Authors
This is a useful study that confirms the value of education in driving vaccination rates. One addition to the study that might be useful is to include a paragraph about the nature of the education. What was discussed? Was it primarily educational or persuasive? How long was the educational effort? Who delivered the education? These are important details to provide some indication of why the intervention was effective. As it stands, there is no reasonable way to determine why the intervention was effective.
Author Response
Comment: This is a useful study that confirms the value of education in driving vaccination rates. One addition to the study that might be useful is to include a paragraph about the nature of the education. What was discussed? Was it primarily educational or persuasive? How long was the educational effort? Who delivered the education? These are important details to provide some indication of why the intervention was effective. As it stands, there is no reasonable way to determine why the intervention was effective.
Reply: Thank you very much for your valuable feedback and detailed insights. We agree. Based on your suggestions, we have incorporated additional details about the educational approach used in our study, drawing on counselling principles from the Italian National Institute of Health (ISS). This included a focus on welcoming, listening, and respecting individual decisions regarding vaccination, aimed at facilitating informed and autonomous choices. We have updated the introduction and the materials and methods sections to reflect these additional details. Once again, we sincerely appreciate your constructive review and for helping us improve the clarity and completeness of our study.
Thank you again for your insightful comments.
Reviewer 4 Report
Comments and Suggestions for Authors
General comment
The paper “Effectiveness of a Counseling Intervention to Increase Vaccination Uptake Among Men Who Have Sex with Men During the Mpox Outbreak” is an interesting article about the effectiveness of a vaccination counseling intervention carried out during the administration of the mpox vaccine. The article is well written and the results may be useful for future recommendations on improving vaccination coverage in vulnerable population groups but some aspects needs editorial review.
Major comments
The analysis of factors associated with the effectiveness of the advice for improving coverage should be carried out with the independent variable "increase in vaccination coverage" not with post-counseling vaccine coverage. Review and change in Abstract, Methods and Results (table 2).
In the Introduction, the background of the effectiveness of counseling to improve vaccination coverage should be commented on and the methods should better explain the counseling carried out.
Specific comments
1) Abstract: review implications in conclusion.
2) Review Introduction and Methods section and explain better the counseling intervention
3) Review Methods and Results. Change table 2 using as dependent variable "increase in vaccination coverage".
4) Discuss the implication of pre-counseling uptake vaccination in the results
5) Discuss better the implication of taking advantage of a public health intervention on a specific problem to improving other (the vaccination uptake in this case).
6) Check the reference style
Comments on the Quality of English LanguageGeneral comment
The paper “Effectiveness of a Counseling Intervention to Increase Vaccination Uptake Among Men Who Have Sex with Men During the Mpox Outbreak” is an interesting article about the effectiveness of a vaccination counseling intervention carried out during the administration of the mpox vaccine. The article is well written and the results may be useful for future recommendations on improving vaccination coverage in vulnerable population groups but some aspects needs editorial review.
Major comments
The analysis of factors associated with the effectiveness of the advice for improving coverage should be carried out with the independent variable "increase in vaccination coverage" not with post-counseling vaccine coverage. Review and change in Abstract, Methods and Results (table 2).
In the Introduction, the background of the effectiveness of counseling to improve vaccination coverage should be commented on and the methods should better explain the counseling carried out.
Specific comments
1) Abstract: review implications in conclusion.
2) Review Introduction and Methods section and explain better the counseling intervention
3) Review Methods and Results. Change table 2 using as dependent variable "increase in vaccination coverage".
4) Discuss the implication of pre-counseling uptake vaccination in the results
5) Discuss better the implication of taking advantage of a public health intervention on a specific problem to improving other (the vaccination uptake in this case).
6) Check the reference style
Author Response
The paper “Effectiveness of a Counseling Intervention to Increase Vaccination Uptake Among Men Who Have Sex with Men During the Mpox Outbreak” is an interesting article about the effectiveness of a vaccination counseling intervention carried out during the administration of the mpox vaccine. The article is well written and the results may be useful for future recommendations on improving vaccination coverage in vulnerable population groups but some aspects needs editorial review.
Reply: Thank you very much for your thoughtful feedback on our paper. We are pleased to hear that you found the article interesting and appreciate your kind words about its potential impact.
Comment: The analysis of factors associated with the effectiveness of the advice for improving coverage should be carried out with the independent variable "increase in vaccination coverage" not with post-counseling vaccine coverage. Review and change in Abstract, Methods and Results (table 2).
Reply: The independent variable in the regression analysis is a binary variable indicating vaccination status after the intervention (0 = non vaccinated; 1 =vaccinated). Therefore it intrinsically means an increase in vaccination coverage, as vaccination coverage cannot diminish. We, however, agree that this could be not clear to the reader. The text and Table 2 title have been amended accordingly.
Comment: In the Introduction, the background of the effectiveness of counseling to improve vaccination coverage should be commented on and the methods should better explain the counseling carried out.
Reply: Thank you, we agree. The introduction has been expanded to include various methods to enhance coverage, addressing hesitancy, and covering both mass communication with the example of COVID and the effectiveness of vaccine counseling.
Comments:
- Abstract: review implications in conclusion.
Reply: Thank you. We have made modifications within the word limit and based on the other suggestions provided on how to enhance the discussion and implications sections.
2) Review Introduction and Methods section and explain better the counseling intervention
Reply: Thank you for your feedback. We have revised the Introduction and Methods section to better explain the counseling intervention. We appreciate your suggestions and agree that clarity on the counseling approach is crucial for understanding our study's methodology.
3) Review Methods and Results. Change table 2 using as dependent variable "increase in vaccination coverage".
Reply: This has been changed (see also our reply above).
4) Discuss the implication of pre-counseling uptake vaccination in the results
Reply: Thank you for your feedback. We have addressed the implication of the pre-counseling uptake vaccination in the discussion. Specifically, we highlighted that the initial coverage rates were low. These initial baseline coverage rates can be found in the tables in the Results section. This context has significant implications for understanding the impact of the counseling intervention, especially in boosting HPV and HAV vaccination rates. We appreciate your suggestion and believe this addition strengthens the discussion.
5) Discuss better the implication of taking advantage of a public health intervention on a specific problem to improving other (the vaccination uptake in this case).
Reply: Thank you very much, we agree that this is an interesting point. In the discussion, we added implications of taking advantage of a public health intervention on a specific problem to improve other outcomes. The experiences of when a subject is vaccinated show that they are more likely to have been vaccinated for something else, and the possible determinants.
6) Check the reference style
Reply: We did it, thank you.
Round 2
Reviewer 1 Report
Comments and Suggestions for Authors
The manuscript has successfully improved and is now ready to be published
Reviewer 2 Report
Comments and Suggestions for Authors
The revised manuscript has adequately and precisely addressed all the comments and concerns raised in the first review
Reviewer 4 Report
Comments and Suggestions for Authors
The authors have adressed most of the topic of the review and have answered the different issues.